# Association between Death or Hospitalization and Observable Variables of Eating and Swallowing Function among Elderly Residents in Long-Term Care Facilities: A Multicenter Prospective Cohort Study

**DOI:** 10.3390/healthcare11131827

**Published:** 2023-06-22

**Authors:** Maaya Takeda, Yutaka Watanabe, Kenshu Taira, Kazuhito Miura, Yuki Ohara, Masanori Iwasaki, Kayoko Ito, Junko Nakajima, Yasuyuki Iwasa, Masataka Itoda, Yasuhiro Nishi, Yoshihiko Watanabe, Masako Kishima, Hirohiko Hirano, Maki Shirobe, Shunsuke Minakuchi, Mitsuyoshi Yoshida, Yutaka Yamazaki

**Affiliations:** 1Gerodontology, Department of Oral Health Science, Faculty of Dental Medicine, Hokkaido University, Sapporo 060-8586, Japan; takedamaaya@gmail.com (M.T.); kenshu-t@med.u-ryukyu.ac.jp (K.T.); kzhtmiura@gmail.com (K.M.); yutaka8@den.hokudai.ac.jp (Y.Y.); 2Tokyo Metropolitan Institute for Geriatrics and Gerontology, Itabashi-ku 173-0015, Japan; yohara@tmig.or.jp (Y.O.); iwasaki@tmig.or.jp (M.I.); h-hiro@gd5.so-net.ne.jp (H.H.); mshirobe@tmig.or.jp (M.S.); 3Department of Public Health and Hygiene, Graduate School of Medicine, University of the Ryukyus, Nishihara 903-0215, Japan; 4Division of Preventive Dentistry, Department of Oral Health Science, Graduate School of Dental Medicine, Hokkaido University, Sapporo 060-8586, Japan; 5Oral Rehabilitation, Niigata University Medical and Dental Hospital, Niigata 951-8520, Japan; k-ito@dent.niigata-u.ac.jp; 6Department of Oral Medicine and Hospital Dentistry, Tokyo Dental College, Ichikawa 272-8513, Japan; jun.k.nakajima@gmail.com; 7Department of Dentistry, Haradoi Hospital, Fukuoka 813-8588, Japan; y_iwasa@haradoi-hospital.com; 8Department of Oral Rehabilitation, Osaka Dental University Hospital, Osaka 573-1144, Japan; m.itoda89@gmail.com; 9Department of Oral and Maxillofacial Prosthodontics, Kagoshima University Graduate School of Medical and Dental Sciences, Kagoshima 890-8544, Japan; shar@dent.kagoshima-u.ac.jp; 10Department of Healthcare Management, Tohoku Fukushi University, Sendai 981-8522, Japan; yoshiw@tfu.ac.jp; 11Department of Dentistry, Wakakusa-Tatsuma Rehabilitation Hospital, Daito 574-0012, Japan; tdbqd500@yahoo.co.jp; 12Gerodontology and Oral Rehabilitation, Graduate School of Medical and Dental Sciences, Tokyo Medical and Dental University, Bunkyo-ku 113-8549, Japan; s.minakuchi.gerd@tmd.ac.jp; 13Department of Dentistry and Oral-Maxillofacial Surgery, School of Medicine, Fujita Health University, Toyoake 470-1192, Japan; mitsuyoshi.yoshida@fujita-hu.ac.jp

**Keywords:** eating, swallowing function, long-term care facility, drooling, halitosis, rinsing, geriatrics, gerodontology, geriatric care

## Abstract

This 1-year multicenter prospective cohort study aimed to determine the association between observable eating and swallowing function factors and outcomes (death/hospitalization or survival) among elderly persons in long-term care insurance facilities in Japan. Baseline assessments of factors, such as language, drooling, halitosis, hypersalivation, tongue movement, perioral muscle function, coughing, respiration after swallowing, rinsing, and oral residue, among others, were conducted. A score of 0 was considered positive, and a score of 1 or 2 was considered negative. Patient age, sex, body mass index, Barthel index, and Clinical Dementia Rating were recorded. The death/hospitalization or survival rates over 1 year were recorded, and patients were allocated into groups depending on the respective outcome (death/hospitalization group or survival group) and baseline characteristics. A total of 986 residents from 32 facilities were included, with 216 in the death/hospitalization group and 770 in the survival group. Language, salivation, halitosis, perioral muscle, coughing, respiration after swallowing, rinsing, and oral residue were significantly associated with the outcomes (*p* < 0.05). Therefore, routine performance of these simple assessments by caregivers may allow early detection and treatment to prevent death, pneumonia, aspiration, and malnutrition in elderly persons.

## 1. Introduction

A recent Japanese study reported that 25.1% and 53.8% of healthy elderly persons in the general community and nursing homes, respectively, have dysphagia [1]. Early detection of signs of incompatibility between the eating and swallowing functions and food forms, prompt referral to a dysphagia specialist, and intake of appropriate food forms can help prevent aspiration, choking, and malnutrition and maintain quality of life (QOL) [2,3].

Video fluorography (VF) and endoscopy (VE) performed by dysphagia specialists are important to evaluate eating and swallowing functions and determine eating patterns [4,5]. However, routinely performing these examinations for patients at medical institutions, nursing homes, and home is difficult [6].

Caregivers (nurses, caregivers, and in some cases, family members) routinely observe the eating and swallowing functions of older patients who require nursing care and whose eating and swallowing functions are gradually declining due to aging (and not due to disease). It is critical to detect signs of incompatibility between function and eating patterns, promptly refer the patient to dysphagia specialists, and improve the patients’ eating patterns. A collaboration between nurses, caregivers, and family members is therefore necessary. Timely diagnosis and interventions for elderly individuals with dysphagia can reduce their risk of pneumonia, malnutrition, choking, and aspiration and help avoid hospitalization, allowing continued home care and lower medical and nursing care costs.

Previously, we conducted a cross-sectional study comparing Japanese elderly patients in long-term care facilities consuming a regular diet to those with dietary adjustment. We hypothesized that food morphology could be screened with simple assessments of eating and swallowing functions that caregivers can routinely perform. The results revealed that simple assessments of swallowing and whether or not a patient gargled, which could easily and routinely be performed by caregivers, were associated with the food form taken by the patient [7]. Next, we hypothesized that simple observational items, such as rinsing, tongue movement, and perioral muscle function, can predict a change from a normal diet (ND) to a dysphagia diet (DD) and conducted a 1-year prospective, multicenter, longitudinal study of residents in long-term care insurance facilities in Japan to identify factors that can predict an adjusted diet. Our findings suggested that tongue movement, perioral muscle function, and mouth rinsing predict a change from ND to DD [8]. It has been shown that a change in food form is associated with weight loss [9]. Since simple observables associated with a change in food form can also be used to screen eating and swallowing function, we hypothesized that poor eating and swallowing functions are associated with hospitalization, survival, and death outcomes. This study was performed to identify simple observables related to the eating and swallowing functions associated with the aforementioned outcomes.

## 2. Materials and Methods

### 2.1. Study Design

This was a multicenter prospective cohort study.

### 2.2. Subjects

Residents of long-term healthcare facilities in Japan were included in the study. We educated 30 members of a specially appointed committee of the Japanese Association of Geriatric Dentistry on the study design and trained participants to ensure the standardization of assessments. Each member then explained the study to the heads and staff of the collaborating long-term care facilities (32 facilities in 17 regions of Japan). A total of 986 residents and their families were informed in writing about the study in September 2019, and written consent was obtained from both participants and their families. A baseline survey was conducted from October 2019 to February 2020. A resurvey of the 32 facilities that had participated in the study during the previous fiscal year was requested in February 2021 in conjunction with consent for study participation. The study design was approved by the ethics committees of the Japanese Society of Gerodontology (2018-1) and the Hokkaido University Faculty of Dental Medicine (2020 No. 4). A transcription survey of the residents who participated in the study the previous year was also conducted (Figure 1).

### 2.3. Survey Items

Before the survey began, study members provided all nurses and dietitians at the facility with training on the evaluation of survey items and standardization of the evaluation criteria. Questionnaires were distributed, and the following surveys were conducted among residents of each facility by the nurses and dietitians responsible for them.

#### 2.3.1. Basic Information

The dietitian in charge recorded the residents’ age, sex, and body mass index (BMI) from the nursing care records. The BMI was graded as follows: 0 ≥18.5 kg/m^2^; 1 <18.5 kg/m^2^ [10].

#### 2.3.2. Daily Living Function and Cognitive Function

The nurse in charge performed a life function assessment using the Barthel Index (BI) [11]. Cognitive function was assessed with the clinical dementia rating (CDR) based on the method described by Morris et al. [12]. A trained specialist made the final judgment regarding the CDR.

#### 2.3.3. Oral Status

The nurse in charge surveyed the oral status based on specific aspects of residents’ dietary patterns collected by the facility staff beforehand. The assessment was explained to the nurse using a manual. An investigator accompanied the nurse to assess four to five residents to standardize the assessment criteria. As in prior studies, a score of 0 was considered a good outcome, while that of >1 was considered poor. The evaluated parameters of oral status and their scores are summarized in Table 1.

Elderly residents requiring long-term care who participated in the study from the fiscal year 2019 were divided into two groups: the combined death/hospitalization group comprising residents who died or were hospitalized for pneumonia, aspiration, dehydration, or malnutrition during the 1-year period (2020–2021) and the survival group comprising the residents not included in the combined death or hospitalization group.

### 2.4. Statistical Analysis

The baseline characteristics of both groups were compared. Participant sex, oral status, BMI, and CDR in the two groups were compared using the Chi-square test; continuous variables were analyzed using the Kolmogorov–Smirnov normality test, and participant age and BI score were compared using the Mann–Whitney U-test.

Because data from multiple institutions were analyzed, a random effect from the institution was confirmed using multi-level analysis. To examine factors associated with the outcome, we performed a multi-level analysis based on whether the patient was dead, hospitalized, or alive 1 year after the baseline survey as the dependent variable and obtained odds ratios (95% confidence interval).

The independent variables were age, sex, BMI [13], BI [14], and CDR [3], which have been reported to be associated with dietary patterns.

The items of the simple assessment were used as explanatory variables and analyzed one by one. Then, multi-level analyses were performed with death, hospitalization, or survival during the year from the baseline survey as the dependent variable. All statistical analyses were performed using SPSS Statistics 26 (IBM, USA) with a significance set at *p* < 0.05.

## 3. Results

Overall, 986 long-term care facility residents (205 men and 781 women; mean age, 86.8 ± 7.9 years) from 32 facilities in 17 regions of Japan participated in the baseline study.

During the 1-year period after the baseline survey, 216 residents (21.9%) either died (n = 141) or were discharged from the facility for hospitalization (n = 75) and were thus included in the death/hospitalization group. The remaining 770 (78.1%) residents were included in the survivor group. The causes of death or hospitalization in the death/hospitalization group are summarized in Figure 1.

The residents in the survivor group had significantly lower age and CDR, comprised more females, and had significantly higher average BMI and BI scores than those of the death/hospitalization group. Additionally, the survivor group had a significantly higher percentage of participants with good results for all items in the brief evaluation except for the left-right asymmetrical movement of the mouth angle (Table 2).

Since data from multiple centers were analyzed, and the random effects of institutions were confirmed and supported using multi-level analysis, we conducted a multi-level analysis with both groups as dependent variables.

The results showed that language, drooling, halitosis, masticatory movement, tongue movement, perioral muscle function, swallowing, coughing, changes in voice quality after swallowing, respiration after swallowing, rinsing, and oral residue were significantly related to each other. In other words, the findings of the simple assessments, except for those of the left-right asymmetric movement of the mouth angle, were significantly associated with the outcome (i.e., death/hospitalization group or survival) (Table 3, Model 1).

Next, multi-level analyses of the explanatory variables (age, sex, BMI, BI, and CDR) with the two groups (survival and death/hospitalization group) as dependent variables revealed significant differences in all explanatory variables (Table 4).

Additionally, when the simple assessment items and explanatory variables were analyzed individually, significant differences were found in language, drooling, halitosis, perioral muscle function, coughing, and coughing, respiration after swallowing, rinsing, and oral residue (Table 3, Model 2).

Significant differences were found in language, drooling, halitosis, perioral muscle function, swallowing, respiration after swallowing, rinsing, and oral residue in both Models 1 and 2.

## 4. Discussion

The results of this study showed that among the simple assessment items of oral status, language, drooling, halitosis, perioral muscle function, swallowing, respiration after swallowing, rinsing, and oral residue were related to the specified outcomes. Additionally, these simple assessments can be conducted by caregivers who are close to the older adult in need of care, such as during the daily meal service and oral care sessions. In a previous cross-sectional study, we found that coughing and rinsing were associated with distinguishing between regular and adjusted diets during simple assessments. A longitudinal study [8] suggested that tongue movement, perioral muscle function, and rinsing could help predict a change from a regular diet to an adjusted diet. Therefore, subjects with poor perioral muscle function and rinsing function are likely to have impaired eating and swallowing functions. In other words, screening for a decline in the swallowing function through simple observation may help predict outcomes.

It has been reported that hyponutrition due to impaired eating and swallowing functions in elderly persons requiring long-term care can significantly impact the intensity of care requirements and life outcomes [15]. Focusing on the factors associated with the outcomes in the present study could help maintain dietary safety among cognitively impaired elderly persons in need of care and may help predict and prevent aspiration, choking, and malnutrition [2] associated with eating and swallowing dysfunction. The frequent use of VE and VF in long-term care insurance facilities is difficult and requires specific settings. Thus, VE [16,17] and VF [18] were not performed in the previous and present studies as they cannot be used to assess daily eating and swallowing function. Therefore, we believe that the simple assessments, which showed significant associations in this study, can be used to effectively screen patients for referral to a specialist. This suggests that the combined results of the previous and present studies may be very useful. In addition, we believe that these simple assessments can be an effective screening tool for patients requiring nursing care and that the assessment findings could indicate the need for further examination, including swallowing VE and contrast-enhanced swallowing examination, or referral to a specialist in case of poor findings.

Swallowing function, xerostomia, oral hygiene, and the number of teeth are significantly associated with mortality risk in elderly persons requiring long-term care, and cumulative deterioration of oral status can significantly increase the risk of mortality [19,20,21,22]. It was previously reported that tension in the masseter muscle on palpation was associated with mortality at 1 year [19]. Significant differences between outcome groups (death or hospitalization, other) in drooling, halitosis, perioral muscle function, swallowing, respiration after swallowing, rinsing, and oral residue were found in the present study. These significant differences are related to either swallowing function or oral hygiene. Further, these results support those of previous studies [19,20,21,22]. The simple assessments used in this study can prove effective as they can be easily and routinely performed by people without specialized knowledge or tools. The researcher explained the simple assessments to the nurses in advance using a manual. The nurses evaluated four to five participants together with the researcher to ensure standardization of the assessment. In other words, valid results can likely be obtained without specialized training, and dissemination of these assessments would be easy and useful.

Among the items of the simple assessments, the one that showed no significant difference between Model 1 and Model 2 was the left-right asymmetric movement of the mouth angle. Symmetric movement of the mouth angle was considered normal, while the asymmetric movement of the mouth angle was considered problematic. Symmetric movements are often difficult in cases of impairment of cognitive function. Therefore, the lack of a significant difference in this parameter between Models 1 and 2 may be attributable to most participants of the present study (89.5%) having a cognitive decline of CDR1 (mild dementia) or higher.

### 4.1. Generalizability

While our prior study included 431 residents who participated in both the 2018 and 2019 surveys, this study examined and analyzed outcomes up to 1 year later for 986 residents who participated in the 2019 survey. There were 394 duplicate residents and 323 residents from nine additional sites in addition to those of the previous study [8]. Thus, although the data on the subjects analyzed are different, both have significant results for similar endpoints, indicative of the reproducibility and validity of the results.

In addition, age, sex, BMI, BI score, and CDR, which were used as explanatory variables, were all significantly different when a multi-level analysis was performed with the survivor group and the death/hospitalization group as the two dependent variables. The same was true in our previous studies [7,8], the findings of which support the present results.

The average age of the participants analyzed in this study was 86.8 years, and the percentage of individuals with cognitive impairment of CDR1 or higher was 89.5%. The average age of nursing home residents in the United States was reportedly 84 years [23]. In a study of nursing homes in South Korea, the average age of residents was 80.7 years, and 85.8% had mild to severe cognitive decline [24]. A study on Japanese long-term care insurance facilities showed an average resident age of 85.2 years, with 91.3% of residents having a cognitive decline of CDR1 or higher [19]. The participants of Japanese studies, including this study, are older and have a higher percentage of cognitively impaired individuals. However, Japan has the highest proportion of elderly people in the world (28.4% of the Japanese population) and also has a large proportion of cognitively impaired people. Considering these factors, the participants of this study can be described as general nursing care insurance facility residents. The present study population may also be considered representative of future residents of long-term care insurance facilities in countries with an aging population. Furthermore, this study was conducted when the incidence of the COVID-19 infection was high. In a previous study conducted at a Japanese long-term care insurance facility, the mortality rate was reported to be 19.6% [25]. Additionally, another previous study in a Japanese long-term care insurance facility reported a mortality rate of 22.4% [19]. The mortality rate observed in the present study was 21.9%. There was no significant difference in the percentage of deaths observed in the present and previous studies. Therefore, the present study population may be representative of the general population of residents in Japanese long-term care insurance facilities.

### 4.2. Limitations

First, the facilities surveyed in this study were those to which members of the Japanese Society of Geriatric Dentistry are affiliated. Therefore, it should be noted that a bias related to the sampling of facilities was present. Second, we did not consider the occurrence of new diseases or exacerbations of comorbidities unrelated to death or hospitalization during the observation period, which may have influenced the outcomes. However, we believe these are highly individualized and infrequent and may have had little impact on the results. We also did not consider the presence or absence of dental treatment, which may affect the findings of the simple assessment. We believe that the effect of this limitation on our findings was minimal, as dental visits were generally discouraged during the study period due to the COVID-19 pandemic [26]. Finally, the current study showed that the findings of the simple assessments of swallowing function, such as swallowing, coughing, changes in voice quality after swallowing, and respiration after swallowing, were associated with the outcomes, although we did not assess swallowing through VE or contrast-enhanced imaging, which are the gold standards for examination of swallowing function. Although it is challenging to perform these examinations in a multicenter study with a large sample size, such as the present study, detailed research involving these examinations should be performed in the future.

## 5. Conclusions

Our results suggest that the simple assessments described here can be performed by caregivers close to elderly persons requiring nursing care without specialized knowledge. The assessment findings were associated with the outcomes of elderly persons requiring nursing care in Japanese long-term care insurance facilities. Therefore, routine performance of these simple assessments by caregivers may allow early detection and treatment to prevent pneumonia, aspiration, malnutrition, and death among elderly persons requiring long-term care and having eating and swallowing disorders. In the future, we shall disseminate the simple assessments verified in the present study to nursing care facilities and further test the effectiveness of these measures.

## Figures and Tables

**Figure 1 healthcare-11-01827-f001:**
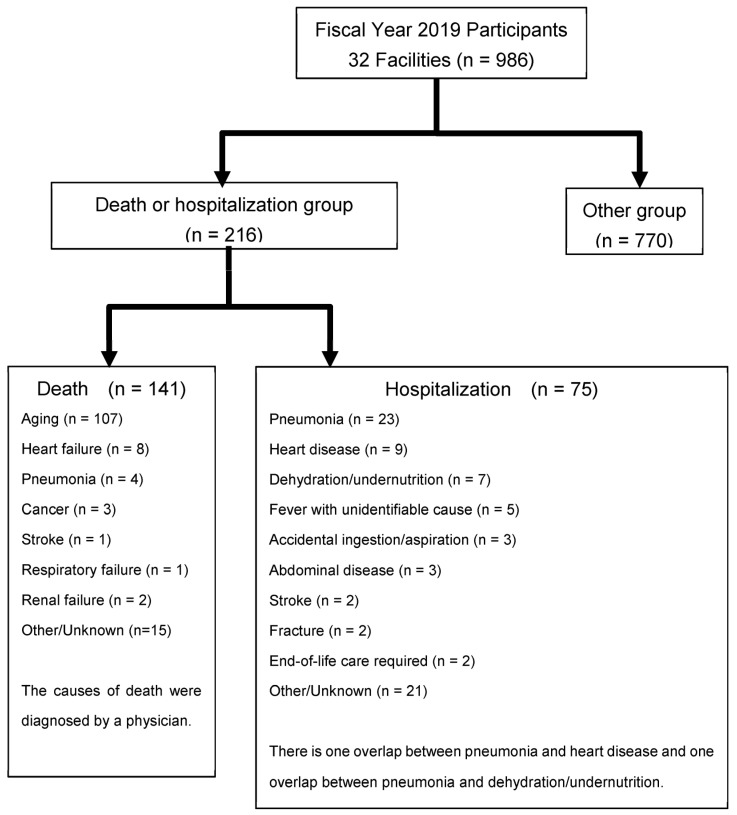
Flow chart of study participation.

**Table 1 healthcare-11-01827-t001:** Oral status parameters and scoring.

Parameter	Scores		
	0	1	2
Language	able to speak	able to speak but with poor articulation	not able to speak
Drooling	no drooling	occasional drooling	always drooling.
Halitosis	no halitosis	a little halitosis	a lot of halitosis
Masticatory movement	has movement	moves when spoken to	almost no movement
Tongue movement	almost complete movement	movement but within a small range	no movement
Perioral muscle function	movement	slight difficulty	no movement
Left-right asymmetric movement of the mouth angle	no	yes	
Swallowing	possible	delayed but possible	
Coughing	no coughing	coughing	
Changes in voice quality after swallowing	no abnormality	abnormality present.	
Respiration after swallowing	no abnormality	shallow and fast breathing	
Rinsing	able to rinse completely	inadequate rinsing	unable to rinse
Oral residue	none	a small amount	present

**Table 2 healthcare-11-01827-t002:** Comparison of characteristics of the study participants.

Variable	Overall (n = 986)				Death or Hospitalization (n = 216)				Other (n = 770)				*p*-Value

	Mean ± SD			Median, [Q1, Q3]	Mean ± SD			Median, [Q1, Q3]	Mean ± SD			Median, [Q1, Q3]	
	*n* (%)				*n* (%)				*n* (%)				
Age	86.8	±	7.9	88.0 [82.0, 92.0]	88.5	±	7.2	89.0 [84.0, 94.0]	86.3	±	8.0	87.0 [82.0, 92.0]	<0.001
Sex (female), n (%)	781		(79.9)		156		(72.9)		625		(81.8)		0.004
Body mass index (<18.5)	597	±	(69.6)		106	±	(56.7)		491	±	(73.2)		<0.001
Barthel Index (Total points)	30.2	±	26.4	25.0 [5.0, 45.0]	20.8	±	23.3	10.0 [0.0, 35.0]	32.9	±	26.7	30.0 [10.0, 50.0]	<0.001
Clinical Dementia Rating (Total points)													
0, 0.5	96		(10.5)		11		(5.6)		85		(11.8)		<0.001
1	127		(13.9)		21		(10.7)		106		(14.8)		
2	231		(25.3)		40		(20.4)		191		(26.6)		
3	460		(50.3)		124		(63.3)		336		(46.8)		
Simple evaluations (oral conditions)													
Language (possible)	672		(68.5)		117		(54.4)		555		(72.5)		<0.001
Drooling (none)	728		(74.4)		132		(61.7)		596		(78.0)		<0.001
Halitosis (none)	571		(58.3)		104		(48.6)		467		(61.0)		<0.001
Masticatory movement (move)	784		(80.2)		151		(70.6)		633		(83.0)		<0.001
Tongue movement (move)	654		(66.9)		110		(51.4)		544		(71.3)		<0.001
Perioral muscle (move)	727		(74.4)		127		(59.6)		600		(78.5)		<0.001
Left-right asymmetric movement of the mouth angle (not)	860		(88.6)		187		(87.8)		673		(88.8)		0.688
Swallowing (not)	765		(79.1)		141		(67.8)		624		(82.2)		<0.001
Coughing (not)	558		(57.2)		87		(40.8)		471		(61.7)		<0.001
Changes in voice quality after swallowing (not)	809		(83.2)		155		(72.8)		654		(86.2)		<0.001
Respiration after swallowing (no abnormality)	933		(95.9)		194		(90.7)		739		(97.4)		<0.001
Rinsing (possible)	529		(53.9)		78		(36.1)		451		(58.9)		<0.001
Oral residue (none)	482		(49.3)		76		(35.3)		406		(53.3)		<0.001

SD, standard deviation.

**Table 3 healthcare-11-01827-t003:** Results of the multi-level analysis of the simple evaluations (oral conditions).

	Model 1					Model 2				
Simple Evaluations (Oral Conditions)	OR		95%CI			OR		95%CI		
Language (1, good; 2, bad)	2.12	**	1.66	-	2.69	1.50	*	1.01	-	2.22
Drooling (1, no; 2, yes)	2.09	**	1.46	-	2.99	1.54	*	1.03	-	2.30
Halitosis (1, no; 2, yes)	1.60	*	1.19	-	2.15	1.49	*	1.09	-	2.04
masticatory movement (1, good; 2, bad)	2.09	**	1.43	-	3.05	1.35		0.85	-	2.13
tongue movement (1, good; 2, bad)	2.19	**	1.63	-	2.94	1.51		0.96	-	2.38
perioral muscle function (1, good; 2, bad)	2.29	**	1.68	-	3.13	1.73	**	1.09	-	2.75
Left-right asymmetric movement of the mouth angle (1, good; 2, bad)	1.25		0.74	-	2.11	1.03		0.61	-	1.75
Swallowing (1, good; 2, bad)	2.09	*	1.29	-	3.38	1.42		0.81	-	2.49
Coughing (1, no; 2, yes)	2.20	**	1.51	-	3.21	1.60	*	1.07	-	2.38
Changes in voice quality after swallowing (1, No abnormality; 2, abnormality)	2.06	*	1.33	-	3.17	1.43		0.93	-	2.22
Respiration after swallowing (1, good; 2, bad)	3.40	**	1.88	-	6.17	2.19	*	1.21	-	3.98
Rinsing (1, possible; 2, impossible)	2.54	**	1.99	-	3.23	2.02	**	1.36	-	3.00
Oral residue (1, no; 2, yes)	1.97	**	1.46	-	2.66	1.66	*	1.13	-	2.43

OR, odds ratio; CI, confidence interval. Model 1: Simple assessment items analyzed one by one as an explanatory variable. Model 2: Simple assessment items and adjustment variables analyzed as explanatory variables (adjustment variables: age, sex, body mass index, Barthel index, and clinical dementia rating). * *p* < 0.05, ** *p* < 0.01.

**Table 4 healthcare-11-01827-t004:** Adjustment variables.

	OR		95%CI		
Age	1.03	**	1.01	-	1.06
Sex (1, male; 2, female)	1.63	*	1.22	-	2.19
Body mass index	0.50	**	0.35	-	0.70
Barthel Index	0.98	**	0.98	-	0.99
Clinical Dementia Rating					
0, 0.5	Reference				
1	1.52		0.69	-	3.37
2	1.33		0.75	-	2.37
3	2.36	**	1.57	-	3.54

CI, confidence interval; OR, odds ratio. * *p* < 0.05. ** *p* < 0.01.

## Data Availability

The data presented in this study are available on request from the corresponding author. The data are not publicly available due to ethicolegal restrictions imposed by the Ethics Committee at the Japanese Society of Gerodontology.

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
