# Peer review of "Association between Death or Hospitalization and Observable Variables of Eating and Swallowing Function among Elderly Residents in Long-Term Care Facilities: A Multicenter Prospective Cohort Study"

_healthcare, 2023, doi:10.3390/healthcare11131827_

Round 1
Reviewer 1 Report
The article is good but needs some revision. Some comments.
It is better to improve the title by adding the words “eating and swallowing function”.
Line 60-61. It is better to add a Reference.
Please add regarding the ethical approval if possible.
Line 116-121. Please add an oral assessment if there is a reference.
Line 277-278. “We believe that dental visits were discouraged during this study period due to the COVID‐19 pandemic and that the effect of this factor on our findings was minimal.” Please add a reference.
https://www.ncbi.nlm.nih.gov/pmc/articles/PMC7654486/
Line 279-280. Please give more clarification for the statement “Finally, we did not perform endoscopic or contrast‐enhanced swallowing, which are the gold standards for assessing swallowing function.”
Author Response
June 14th, 2023
Dear Reviewer:
On behalf of the co-authors, I thank you for the opportunity to revise our manuscript titled “Association Between Death or Hospitalization and Observable Variables of Eating and Swallowing Function among Elderly Residents in Long-Term Care Facilities: A Multicenter Prospective Cohort Study” (ID: healthcare-2233232). The reviewers’ comments guided us significantly and helped us improve our manuscript.
We have carefully addressed all the reviewers’ comments in our revised manuscript. We have also verified that none of the referenced papers have been retracted. The main corrections and point-by-point responses to the reviewers’ comments are attached herewith.
We hope that our responses and revisions adequately address the reviewers’ concerns and that the revised manuscript now meet the standards for publication in your esteemed journal. We look forward to hearing from you.
Sincerely,
Maaya Takeda
Gerodontology, Department of Oral Health Science, Faculty of Dental Medicine, Hokkaido University
Nishi-7, Kita-13, Kita-ku, Sapporo, Hokkaido 060-8586, Japan
Tel: 81-11-706-4582
Fax: 81-11-706-4582
Email: takedamaaya@den.hokudai.ac.jp
1Comments and Suggestions for Authors
The article is good but needs some revision. Some comments.
It is better to improve the title by adding the words “eating and swallowing function”.
→Thank you for this encouraging comment. The title has been revised in accordance with your suggestion.
Line 60-61. It is better to add a Reference.
→ Thank you for this suggestion. We have now added an appropriate reference to support this statement (Page 2, Line 79).
Please add regarding the ethical approval if possible.
→ Thank you. This declaration has been provided on Page 10, Lines 335–339 in keeping with the journal guidelines.
Line 116-121. Please add an oral assessment if there is a reference.
→ We used the same oral assessment method as that used in our previous studies. We have now cited the following references:
https://link.springer.com/article/10.1007/s00455-022-10440-6
https://www.mdpi.com/1660-4601/19/11/6586
Line 277-278. “We believe that dental visits were discouraged during this study period due to the COVID‐19 pandemic and that the effect of this factor on our findings was minimal.” Please add a reference.
https://www.ncbi.nlm.nih.gov/pmc/articles/PMC7654486/
→ Thank you for this suggestion. We have now added an appropriate reference to support this statement (Page 9, Line 300).
Line 279-280. Please give more clarification for the statement “Finally, we did not perform endoscopic or contrast‐enhanced swallowing, which are the gold standards for assessing swallowing function.”
→Thank you. In accordance with this suggestion, we have revised the text for clarity (Page 9, Lines 301–305).
Reviewer 2 Report
Takeda et al. shared a multicenter prospective cohort study to determine the relationship between observable eating and swallowing function factors and outcomes (death/hospitalization or survival) among elderly persons in long‐term care insurance facilities in Japan. This will add one more article to the current literature. Overall, the article is written well in a well-elaborated manner with relevant references. Some of the specific comments are listed below.
Page 1, line 32: Can the author consider including the reference for the cohort study if it was published?
Page 3, line 102: The BMI was graded as follows: 0, ≥18.5 kg/m2; 1, <18.5 kg/m2 [8].
The author can rewrite this for clear understanding, e.g., 0 - ≥18.5 kg/m2.
Why 18.5 kg/m2 was chosen for grading?
Page 4, line 122: Can the author consider writing it in a table format for clear understanding?
Table 2: The author should consider explaining what *, and ** denotes.
Page 8, line 255: … having a cognitive decline of CDR1 or higher.
Can the author consider rewriting this for better understanding? Also, what is CDR1?
Author Response
June 14th, 2023
Dear Reviewer:
On behalf of the co-authors, I thank you for the opportunity to revise our manuscript titled “Association Between Death or Hospitalization and Observable Variables of Eating and Swallowing Function among Elderly Residents in Long-Term Care Facilities: A Multicenter Prospective Cohort Study” (ID: healthcare-2233232). The reviewers’ comments guided us significantly and helped us improve our manuscript.
We have carefully addressed all the reviewers’ comments in our revised manuscript. We have also verified that none of the referenced papers have been retracted. The main corrections and point-by-point responses to the reviewers’ comments are attached herewith.
We hope that our responses and revisions adequately address the reviewers’ concerns and that the revised manuscript now meet the standards for publication in your esteemed journal. We look forward to hearing from you.
Sincerely,
Maaya Takeda
Gerodontology, Department of Oral Health Science, Faculty of Dental Medicine, Hokkaido University
Nishi-7, Kita-13, Kita-ku, Sapporo, Hokkaido 060-8586, Japan
Tel: 81-11-706-4582
Fax: 81-11-706-4582
Email: takedamaaya@den.hokudai.ac.jp
2Comments and Suggestions for Authors
Takeda et al. shared a multicenter prospective cohort study to determine the relationship between observable eating and swallowing function factors and outcomes (death/hospitalization or survival) among elderly persons in long‐term care insurance facilities in Japan. This will add one more article to the current literature. Overall, the article is written well in a well-elaborated manner with relevant references. Some of the specific comments are listed below.
Page 1, line 32: Can the author consider including the reference for the cohort study if it was published?
→ The term “cohort study” at this instance refers to the present study. We have used it to describe the study design in the Abstract.
Page 3, line 102: The BMI was graded as follows: 0, ≥18.5 kg/m2; 1, <18.5 kg/m2 [8].
The author can rewrite this for clear understanding, e.g., 0 - ≥18.5 kg/m2.
Why 18.5 kg/m2 was chosen for grading?
→ Thank you. We have added the relevant citation to support the use of a cut-off point of 18.5 kg/m2 for BMI (Page 4, Lines 119–120).
Page 4, line 122: Can the author consider writing it in a table format for clear understanding?
→ We have expressed the information in a table-like format in keeping with this suggestion (Table 1, Page 4).
Table 2: The author should consider explaining what *, and ** denotes.
→ We have clarified the symbols (* p < 0.05, ** p < 0.01) in Table 3 (originally Table 2) footnotes (Page 7, Line 186).
Page 8, line 255: … having a cognitive decline of CDR1 or higher.
Can the author consider rewriting this for better understanding? Also, what is CDR1?
→CDR1 indicates mild dementia. The text has been revised for clarity (Page 8, Lines 252–255).
Reviewer 3 Report
This is an interesting and well prepared research protocol and presentation. In essence the project tests whether or not a simple, practical assessment of oral health function can be used as a suitable tool to predict serious consequences for elderly long-term care residents. The visual assessment tool would offer an alternative estimate of dysphasia to the high technology options available such as video-fluorography (VF) and video endoscopy (VE), and could be used by trained nurses and caregivers within the institution.
The study comprised 986 residents drawn from 32 institutions. Trained examiners carried out oral health assessments of eating and swallowing functions some 13 different fields (drooling, coughing, language etc) and collected general health and demographic information on the residents, including assessment of activities of daily living and BMI. Each of the oral variables were scored on a 0 (good) to 2 (poor) scale. Mortality and hospitalization rates were collected 12-months following base-line data collection and each resident categorized in either a "survival" or death/hospitalization group for bi and multi variant analysis. The methodology and statistical analyses were sound and clearly reported. Tables and figures were appropriate and clearly labelled. The Discussion was adequate and the conclusions concise and related clearly to the findings. Overall the research demonstrated that the use of a simple series of observable characteristics such as; drooling, halitosis, perioral muscle function, swallowing, rinsing, and oral residue were associated more frequently with subsequent death or hospitalization of elderly care residents. The study did not make a direct comparison between the use of the observation tool and VF or VE. This limitation was noted, but the authors could such how future research may be undertaken to compare the two approaches (high tech vs low tech).
There are minor editorial issues that could be addressed. For example, it may be more appropriate to use the term "association" rather than "relationship" throughout the text. The use of the term study objectives rather than "hypotheses" could be considered in the Introduction. The description of cause of death as "aging" is very broad and needs a degree of clarification - perhaps in the discussion under limitations? Similarly, what were the reasons for two facilities not choosing to participate at the 12-month follow-up? There are a few areas where wording could be improved - see line 245 and 263 where it appears that words are missing in sentence construction.
Author Response
June 14th, 2023
Dear Reviewer:
On behalf of the co-authors, I thank you for the opportunity to revise our manuscript titled “Association Between Death or Hospitalization and Observable Variables of Eating and Swallowing Function among Elderly Residents in Long-Term Care Facilities: A Multicenter Prospective Cohort Study” (ID: healthcare-2233232). The reviewers’ comments guided us significantly and helped us improve our manuscript.
We have carefully addressed all the reviewers’ comments in our revised manuscript. We have also verified that none of the referenced papers have been retracted. The main corrections and point-by-point responses to the reviewers’ comments are attached herewith.
We hope that our responses and revisions adequately address the reviewers’ concerns and that the revised manuscript now meet the standards for publication in your esteemed journal. We look forward to hearing from you.
Sincerely,
Maaya Takeda
Gerodontology, Department of Oral Health Science, Faculty of Dental Medicine, Hokkaido University
Nishi-7, Kita-13, Kita-ku, Sapporo, Hokkaido 060-8586, Japan
Tel: 81-11-706-4582
Fax: 81-11-706-4582
Email: takedamaaya@den.hokudai.ac.jp
3Comments and Suggestions for Authors
This is an interesting and well prepared research protocol and presentation. In essence the project tests whether or not a simple, practical assessment of oral health function can be used as a suitable tool to predict serious consequences for elderly long-term care residents. The visual assessment tool would offer an alternative estimate of dysphasia to the high technology options available such as video-fluorography (VF) and video endoscopy (VE), and could be used by trained nurses and caregivers within the institution.
The study comprised 986 residents drawn from 32 institutions. Trained examiners carried out oral health assessments of eating and swallowing functions some 13 different fields (drooling, coughing, language etc) and collected general health and demographic information on the residents, including assessment of activities of daily living and BMI. Each of the oral variables were scored on a 0 (good) to 2 (poor) scale. Mortality and hospitalization rates were collected 12-months following base-line data collection and each resident categorized in either a "survival" or death/hospitalization group for bi and multi variant analysis. The methodology and statistical analyses were sound and clearly reported. Tables and figures were appropriate and clearly labelled. The Discussion was adequate and the conclusions concise and related clearly to the findings. Overall the research demonstrated that the use of a simple series of observable characteristics such as; drooling, halitosis, perioral muscle function, swallowing, rinsing, and oral residue were associated more frequently with subsequent death or hospitalization of elderly care residents. The study did not make a direct comparison between the use of the observation tool and VF or VE. This limitation was noted, but the authors could such how future research may be undertaken to compare the two approaches (high tech vs low tech).
There are minor editorial issues that could be addressed. For example, it may be more appropriate to use the term "association" rather than "relationship" throughout the text. The use of the term study objectives rather than "hypotheses" could be considered in the Introduction. The description of cause of death as "aging" is very broad and needs a degree of clarification - perhaps in the discussion under limitations? Similarly, what were the reasons for two facilities not choosing to participate at the 12-month follow-up? There are a few areas where wording could be improved - see line 245 and 263 where it appears that words are missing in sentence construction.
→We have revised the term "relationship" to "association" as suggested (Page 1, Line 35).
Aging was diagnosed as the cause of death by a physician. We did not investigate the basis for this diagnosis. We have mentioned in Figure 1 that the causes of death were diagnosed by a physician.
The reason two facilities did not choose to participate at the 12-month follow-up is the increased workload following the spread of the novel coronavirus. Therefore, we believe that this did not impact our study findings. Regarding lines 245 and 263, we request you to specify the exact text that requires revision. We would be happy to incorporate the required language-related changes in accordance with your suggestions.
Reviewer 4 Report
Manuscript titled "Association Between Death and Observable Oral Health Assessment Variables of Elderly Residents in Long-Term Care Facilities: A Multicenter Prospective Cohort Study" showcases an interesting method to assess the risk of death and hospitalization with observable oral variables.
All in all the manuscript is well written, but there are few things that I would like authors to consider:
1. Please consider revising the title of manuscript since outcome also included hospitalization. For example: "Association Between Death or Hospitalization and Observable Oral Health...."
2. Introduction is relatively short and it does not provide sufficient background for the study. Furthermore, majority of the introduction is designated for previous works of the same research group. I recommend authors to extend introduction with more background information on the topic and reduce the text about their previous work.
3. Why death and hospitalization were group together? I think it would increase the value of the manuscript and improve it if these variables were also analyzed/reported individually.
4. Table 3 is missing footnotes for "*" and "**".
5. On line 219: Open which outcomes you mean.
6. On lines 239-240: Do you mean significant differences between outcome groups (death/hospitalization vs. survivors)? It is not clear.
7. Can these result be generalized to other regions?
Author Response
June14th, 2023
Dear Reviewer:
On behalf of the co-authors, I thank you for the opportunity to revise our manuscript titled “Association Between Death or Hospitalization and Observable Variables of Eating and Swallowing Function among Elderly Residents in Long-Term Care Facilities: A Multicenter Prospective Cohort Study” (ID: healthcare-2233232). The reviewers’ comments guided us significantly and helped us improve our manuscript.
We have carefully addressed all the reviewers’ comments in our revised manuscript. We have also verified that none of the referenced papers have been retracted. The main corrections and point-by-point responses to the reviewers’ comments are attached herewith.
We hope that our responses and revisions adequately address the reviewers’ concerns and that the revised manuscript now meet the standards for publication in your esteemed journal. We look forward to hearing from you.
Sincerely,
Maaya Takeda
Gerodontology, Department of Oral Health Science, Faculty of Dental Medicine, Hokkaido University
Nishi-7, Kita-13, Kita-ku, Sapporo, Hokkaido 060-8586, Japan
Tel: 81-11-706-4582
Fax: 81-11-706-4582
Email: takedamaaya@den.hokudai.ac.jp
4Comments and Suggestions for Authors
Manuscript titled "Association Between Death and Observable Oral Health Assessment Variables of Elderly Residents in Long-Term Care Facilities: A Multicenter Prospective Cohort Study" showcases an interesting method to assess the risk of death and hospitalization with observable oral variables.
All in all the manuscript is well written, but there are few things that I would like authors to consider:
- Please consider revising the title of manuscript since outcome also included hospitalization. For example: "Association Between Death or Hospitalization and Observable Oral Health...."
→Thank you. The title has been revised as suggested.
- Introduction is relatively short and it does not provide sufficient background for the study. Furthermore, majority of the introduction is designated for previous works of the same research group. I recommend authors to extend introduction with more background information on the topic and reduce the text about their previous work.
→ In the original Introduction, we had described previous results from our research group, which formed the basis for this study. In the revised manuscript, we have provided more background related to our previous studies (Page 2, Lines 63–72).
- Why death and hospitalization were group together? I think it would increase the value of the manuscript and improve it if these variables were also analyzed/reported individually.
→In this study, we did not follow the course of events after hospitalization due to illness. However, many of the elderly requiring long-term care who were hospitalized and could not return to a long-term care facility died. We therefore combined death and hospitalization.
- Table 3 is missing footnotes for "*" and "**".
→We have clarified the symbols (* p < 0.05, ** p < 0.01) in Table 4 (originally Table 3) footnotes (Page 7, Lines 194–195).
- On line 219: Open which outcomes you mean.
→The outcomes here refer to death, hospitalization, and survival (Page 8, Line 216).
- On lines 239-240: Do you mean significant differences between outcome groups (death/hospitalization vs. survivors)? It is not clear.
→ Thank you for bringing this to our attention. We meant significant difference between outcome groups (death and hospitalization vs. survivors). The text has been revised to accordingly (Page 8, Lines 237–238).
- Can these result be generalized to other regions?
→ We have made additions regarding the generalizability of our findings to other regions (Page 9, Lines 268–289).
Round 2
Reviewer 4 Report
The authors have provided sufficient responses and changes into their manuscript.